

# The effect on twinning rate of transferring double vitrified-warmed embryos in women of advanced reproductive age: a retrospective study

Yamei Xue[1,2], Kun Li[3] and Songying Zhang[1,2]

[1] Assisted Reproduction Unit, Department of Obstetrics and Gynecology, Sir Run Run Shaw Hospital, School of Medicine, Zhejiang University, Hangzhou, Zhejiang, China
[2] Key Laboratory of Reproductive Dysfunction Management of Zhejiang Province, Hangzhou, China
[3] Department of Reproductive Physiology, Zhejiang Academy of Medical Sciences, Hangzhou, Zhejiang, China

Corresponding author
Songying Zhang,
zhangsongying@zju.edu.cn

## ABSTRACT

Twin pregnancies are associated with greater risk of neonatal morbidity and mortality than a singleton. This study was performed to investigate the twin pregnancy rate when two vitrified-warmed embryos are transferred in women of advanced reproductive age (≥35 years at the time of oocyte retrieval) and to evaluate the implications of findings in selecting candidates for elective single embryo transfer (eSET). A retrospective analysis of data which included 2,038 women aged 35–45 years, who underwent vitrified-warmed double embryo transfer (DET), from January 2013 to December 2016 was undertaken. Pregnancy and twin rates were estimated after stratifying by prognostic profile. The twin pregnancy rate was lower in women with poor prognosis (12/96, 12.5%) as compared with that in women with favorable prognosis (102/374, 27.3%) and average prognosis (78/346, 22.5%) with significant differences ($P < 0.05$). The twin rate for women with favorable prognosis was 29.2% (70/240) in the cycles of women aged 35–37 years, 26.8% (26/97) in the cycles of women aged 38–40 years and 16.2% (6/37) in the cycles of women >40 years. The twin rate for women with average prognosis was 25.8% (51/198) in the cycles of women aged 35–37 years, 22.0% (22/100) in the cycles of women aged 38–40 years and 10.4% (5/48) in the cycles of women >40 years. The twin rate for women with poor prognosis was 15.3% (9/59) in the cycles of women aged 35–37 years, 10.3% (3/29) in the cycles of women aged 38–40 years and 0% (0/8) in the cycles of women >40 years. From these results, it was concluded that women with a favorable or average prognosis have a high risk of twin pregnancies. The finding can be used to guide future practice: that is, performing eSET in women with favorable or average prognosis and DET in women with poor prognosis.

Subjects Gynecology and Obstetrics, Women's Health
Keywords Twin pregnancy, Double embryo transfer, Advanced age

## INTRODUCTION

Since the widespread introduction of assisted reproductive techniques (ART) in the last decades, the transfer of more than one embryo with the aim of maximizing pregnancy

rates has led to an increase in the rate of twin pregnancies (*Vlachadis, Vrachnis & Economou, 2014*). Twin gestations are more likely to have adverse maternal and neonatal outcomes compared to singletons, including gestational hypertension, premature births, intrauterine growth retardation, and low birth weight (*Saccone et al., 2019*; *Young & Wylie, 2012*). Twin gestation also compound the health threats to women with advanced age, who have a three-time higher risk of pregnancy-related mortality than do younger counterparts (*MacKay, Berg & Atrash, 2001*). Ideally, the goal of ART is to achieve a singleton gestation (*Styer et al., 2016*; *Thurin et al., 2004*).

Elective single embryo transfer (eSET) is encouraged, as is the subsequent reduction in twin pregnancies. Age is a primary clinic criterion when considering a woman's candidacy for eSET. *Martin et al. (2016)* specified the strongest negative predictor of double embryo implantation was age ≥35 years compared to age <35 years after double embryo transfer (DET). *Kim et al. (2015)* reported maternal age ≤35 years was the cutoff value to predict twin pregnancy following DET. Therefore, recommendations for eSET have focused mainly on women aged <35 years who have a favorable prognosis (*Thurin et al., 2004*). Increasing age is associated with a lower chance of multiple pregnancies (*Martin et al., 2016*). For women with over 35 years of age, transferring two embryos is still their first choice, even with a favorable prognosis, in order to maximize the chance of pregnancy with a single treatment. However, according to a large study performed in the USA, women of over 35 years of age had up to 22.1% of double embryo implantation in the favorable prognosis group in DET cycles (*Martin et al., 2016*). A report from Finland estimated the twin pregnancy rate to be 17.7%, when two embryos were transferred in women aged 36–39 years (*Veleva et al., 2006*). In the case that two embryos are transferred and both are implanted, there may be a missed chance for a successful eSET.

In China, the eSET policy has gradually been well accepted by women aged ≤35 years with a good prognosis without provision of law, and practice has led to a decline in the number of twin pregnancy after IVF (*Yang et al., 2018*). However, for women older than 35 years, data from existing studies are limited and have not been able to provide robust evidence on DET or eSET. The present study was therefore carried out to evaluate the outcome of DET in older Chinese women (aged ≥35 years at the time of oocyte retrieval). Given our large sample, we were able to stratify our population by prognosis based on the number of previous cycles and embryo quality, to better assess twin pregnancy rate.

# MATERIALS AND METHODS

## Subjects

In early 2012, a decision was made at our center to limit patients to the transfer of a maximum of two embryos, to control for a multiple pregnancy rate. This retrospective study analyzed the data of 2,038 women ≥35 years at the time of oocyte retrieval, who underwent transfer of vitrified-warmed double embryos at the Reproductive Medicine Center, Sir Run Run Shaw Hospital, China between January 2013 and December 2016. Each patient contributed only one cycle. All of the transferred embryos were at the Day 3 stage. Exclusion criteria included cycles involving mixed fresh/thawed embryo transfer,

**Table 1 The definition of favorable, average and poor prognosis.**

| Prognosis | Definition |
|---|---|
| Favorable | First embryo transfer and two good-quality embryos transferred |
| | A previous successful embryo transfer and ≥1 good-quality embryos transferred |
| Average | First embryo transfer and ≤1 good-quality embryos transferred |
| | ≥1 failed embryo transfer and ≥1 good-quality embryos transferred |
| | A previous successful embryo transfer and no good-quality embryos transferred |
| Poor | ≥1 failed embryo transfer and no good-quality embryos transferred |

oocyte donation and those in which embryos were cryopreserved by slow freezing. Reproductive Medical Ethics Committee of Sir Run Run Shaw Hospital, College of Medicine, Zhejiang University granted Ethical approval to carry out the study within its facilities (Ethical Application Ref: SRRSHRMEC2017005). Written informed consent was obtained from each participating couple.

The prognostic profiles (i.e., favorable/average/poor) were based on two important predictive factors for pregnancy with IVF: number of previous cycles and embryo quality (*Practice Committee of the American Society for Reproductive Medicine, Practice Committee of the Society for Assisted Reproductive Technology, 2009*). Since maternal age is the most important factor, we divide age into the three groups (35–37, 38–40 and >40 years), to further evaluate the role that maternal age played in twin pregnancy rates. The definition of favorable, average and poor prognosis was shown in Table 1. The favorable prognosis population was defined as women aged ≥35 years who (1) underwent their first cycle of embryo transfer and had two good-quality embryos transferred (2) had a previous successful embryo transfer and ≥1 good-quality embryos transferred. The average prognosis population was defined as women who aged ≥35 years who (1) did not have two good-quality embryos transferred in the first cycle (2) failed to get pregnant in their first transfer cycle and had at least one good-quality embryos transferred, or (3) had a previous successful embryo transfer and no good-quality embryos transferred. Women with a poor prognosis were women aged ≥35 years who had ≥1 failed embryo transfer and no good-quality embryos transferred.

## IVF procedure

The ovarian stimulation and IVF procedures have been previously described (*Xue et al., 2014*, *2018*). Briefly, embryo culture with sequential media was carried out as follows: fertilization was performed in fertilization medium (G-IVF; Vitrolife Sweden AB, Sweden) containing 10% serum substitute supplement (SSS; Irvine Scientific, Santa Ana, CA, USA). The following morning, the oocytes were individually placed in microdrops in cleavage medium (G-1; Vitrolife Sweden AB, Sweden) with 12% SSS and incubated in incubators at 37 °C with 6% $CO_2$, 5% $O_2$ and 89% $N_2$ for 48 h before being frozen. Morphological assessment of embryos was based on the number of blastomeres and the proportion of cytoplasmic fragments and the uniformity of blastomeres (*Xue et al., 2014*).

## Vitrification and warming protocols

Vitrification-warming procedures were performed using the Cryotop cryopreservation technologies (Kitazato; BioPharma Co. Ltd., Fuji City, Japan). The thawed embryos cultured in G-2 medium containing 12% SSS for 2 h prior to transfer. Embryos were defined as surviving if they had with ≥50% of intact blastomeres immediately after warming. The presence of damaged blastomeres was recorded after warming.

## Endometrial preparation

Warming embryos were transferred in natural or hormone replacement treatment (HRT) cycles as described previously (*Xue et al., 2018*). For women with regular menstrual cycles, warmed embryos were transferred in natural cycles. For women with irregular ovulation or anovulatory cycles, HRT was used. Luteal phase support was provided with intramuscular injections of progesterone 80 mg (Xianju Pharmaceutical Factory, Zhejiang, China) from 3 days before ET until a viable intrauterine pregnancy on ultrasound examination was observed.

## Outcomes

For each transfer, this study observed whether or not DET resulted in a clinical pregnancy and, if so, in a singleton or a dizygotic twinning. Clinical pregnancy was defined by the observation of a gestational sac with or without fetal heartbeat on ultrasound evaluation 35 days after embryo transfer. The number of sacs was taken as the number of successful implantations. Live birth referred to the birth of one or more live infants regardless of the duration of pregnancy.

## Statistical analysis

For the statistical analyses, the statistical program Statistical Package for Social Sciences version 20.0 for Windows (SPSS, Chicago, IL, USA) was used. The distributions of continuous variables were evaluated by Kolmogorov Smirnov test. The variables were compared with one way analysis of variance (ANOVA) test and the nonparametric Kruskal–Wallis test depending on normal or non-normal distribution, respectively. The continuous variables were presented as mean ± standard deviation (SD). Categorical variables were presented as $n$ (%) and compared with the Pearson's Chi-squared or Fisher exact test on the basis of sample size. Post hoc pairwise comparisons were performed using the Bonferroni correction. The result was considered significant if the $P$-value was <0.05.

## RESULTS

During the study period, 2,038 women at least 35 years old underwent vitrified-warmed DET. They were divided according to prognosis to 812 cycles performed on women with favorable prognosis, 906 cycles performed on women with average prognosis and 320 cycles performed on women with poor prognosis.

The patients' characteristics are summarized for each prognostic category in Table 2. The most common infertility cause was male factor infertility for patients with favorable

**Table 2  Patient characteristics and outcomes of the included cycles with double embryo transfer.**

| Variable | Included cycles (n = 2038) | Prognosis | | | P-value[a] |
|---|---|---|---|---|---|
| | | Favorable (n = 812) | Average (n = 906) | Poor (n = 320) | |
| Maternal age[b,d] | 37.8 ± 2.7 | 37.6 ± 2.6 | 38.0 ± 2.8 | 37.9 ± 2.7 | 0.007 |
| Maternal BMI | 21.2 ± 2.5 | 21.1 ± 2.5 | 21.2 ± 2.5 | 21.2 ± 2.4 | 0.957 |
| Cause of infertility | | | | | |
| Tubal pathology | 682 (33.5) | 251 (30.9) | 322 (35.5) | 109 (34.1) | 0.124 |
| Endometriosis | 195 (9.6) | 76 (9.5) | 89 (9.8) | 30 (9.4) | 0.940 |
| Male factor[c,d] | 713 (35.0) | 265 (32.6) | 316 (34.9) | 132 (41.3) | <0.001 |
| Unexplained[b,d] | 230 (11.3) | 114 (14.0) | 90 (9.9) | 26 (8.1) | 0.004 |
| Others[b,d] | 218 (10.7) | 106 (13.1) | 89 (9.8) | 23 (7.2) | 0.008 |
| Type of infertility (%) | | | | | 0.069 |
| Primary infertility | 1,665 (81.7) | 647 (79.7) | 760 (83.9) | 258 (80.6) | |
| Secondary infertility | 373 (18.3) | 165 (20.3) | 146 (16.1) | 62 (19.4) | |
| Fertilization method (%)[b,d] | | | | | 0.006 |
| IVF | 1,257 (61.7) | 534 (65.8) | 541 (59.7) | 182 (56.9) | |
| ICSI | 781 (38.3) | 278 (34.2) | 365 (40.3) | 138 (43.1) | |
| Length of embryo cryopreservation (month) | 5.7 ± 3.8 | 5.5 ± 3.9 | 5.8 ± 3.7 | 5.8 ± 4.1 | 0.414 |
| Survival rate (%) | 4,119/4,213 (97.8) | 1,642/1,681 (97.7) | 1,832/1,874 (97.8) | 645/658 (98.0) | 0.879 |
| Damaged blastomere (%) | | | | | 0.367 |
| Yes | 221 (10.8) | 80 (9.9) | 108 (11.9) | 33 (10.3) | |
| No | 1,817 (89.2) | 732 (90.1) | 798 (88.1) | 287 (89.7) | |
| Endometrial thickness (mm) | 9.1 ± 1.6 | 9.1 ± 1.6 | 9.1 ± 1.6 | 8.9 ± 1.7 | 0.136 |
| Endometrial preparation (%) | | | | | 0.367 |
| Natural cycles | 126 (6.2) | 41 (5.0) | 65 (7.2) | 20 (6.3) | |
| HRT cycles | 1,912 (93.8) | 771 (95.0) | 841 (92.8) | 300 (93.7) | |
| No. of good-quality embryos transferred[b,c,d] | 1.1 ± 0.9 | 1.9 ± 0.1 | 0.8 ± 0.6 | 0 | <0.001 |
| Clinical pregnancy[b,c,d] | 816 (40.0) | 374 (46.1) | 346 (38.2) | 96 (30.0) | <0.001 |
| Implantation rate[b,c,d] | 1,008/4076 (24.7) | 470/1,624 (28.9) | 430/1,812 (23.7) | 108/640 (16.9) | <0.001 |
| Twin pregnancy[c,d] | 192 (23.5) | 102/374 (27.3) | 78/346 (22.5) | 12/96 (12.5) | 0.008 |
| Live birth[b,c,d] | 734 (36.0) | 335 (41.3) | 318 (35.1) | 81 (25.3) | <0.001 |

Notes:
[a] P-value for global comparison between the three groups.
[b] Significant comparison between women with favorable and average prognosis.
[c] Significant comparison between women with average and poor prognosis.
[d] Significant comparison between women with favorable and poor prognosis.

and poor prognosis and tubal pathology infertility for women with average prognosis. The differences for patients' age, fertilization method, and number of good-quality embryos transferred among the three groups were statistically significant, but the maternal BMI, type of infertility, length of embryo cryopreservation, damaged blastomere, embryo survival rate, endometrial thickness, and endometrial preparation were similar among the three groups of patients.

In the cycles of women with a favorable prognosis, the clinical pregnancy rate was 46.1% (374/812) per transfer with twin pregnancy rate of 27.3% (102/374) and the live birth rate of 41.3% (335/812). In the cycles of women with an average prognosis, the clinical pregnancy rate was 38.2% (346/906), the twin pregnancy rate was 22.5% (78/346) and the live birth rate was 35.1% (318/906). The rates of clinical pregnancy, twin pregnancy and live birth for women with poor prognosis were 30.0% (96/320),12.5% (12/96) and 25.3% (81/320), respectively. The differences for clinical pregnancy, twin and live birth rates among the three groups were statistically significant ($P < 0.01$).

The patients were subdivided by the women's age into three groups: 35–37, 38–40 and >40 years. The pregnancy, twin and live birth rates for women with favorable prognosis were 49.3% (240/487), 29.2% (70/240) and 44.6% (217/487) in the cycles of women aged 35–37 years, 43.9% (97/221), 26.8% (26/97) and 38.9% (86/221) in the cycles of women aged 38–40 years, 35.6% (37/104), 16.2% (6/37) and 30.8% (32/104) in the cycles of women >40 years. The pregnancy, twin and live birth rates for women with average prognosis were 41.0% (198/483), 25.8% (51/198) and 38.3% (185/483) in the cycles of women aged 35–37 years, 38.2% (100/262), 22.0% (22/100) and 34.7% (91/262) in the cycles of women aged 38–40 years, 29.8% (48/161), 10.4% (5/48) and 26.1% (42/161) in the cycles of women >40 years. The pregnancy, twin and live birth rates for women with poor prognosis were 36.0% (59/164), 15.3% (9/59) and 31.1% (51/164) in the cycles of women aged 35–37 years, 29.6% (29/98), 10.3% (3/29) and 24.5% (24/98) in the cycles of women aged 38–40 years, 13.8% (8/58), 0% (0/8) and 10.3% (6/58) in the cycles of women >40 years. The data are reported in Table 3.

## DISCUSSION

The relaxation of the one child policy in 2013 in China and introduction of the two child policy in 2015 has led to more and more women of advanced reproductive age seeking assisted reproduction (*Liang et al., 2018*). One of the major unsolved issues in IVF is optimizing pregnancy rates while limiting multiple pregnancy rates in older women (≥35 years) with favorable or average prognosis, such as women who have one or more good-quality embryos available in the first transfer cycle. This retrospective evaluation of DET based on the prognostic profile of the individual patient shows that in women with favorable and average prognosis DET results in good pregnancy rates, but at the expense of high twin pregnancy rates. Women with a poor prognosis undergoing DET have reasonable clinical pregnancy rate with a limited twin pregnancy rate (12.5%).

It is well known that maternal age is the most significant indicator for double embryo implantation following DET (*Kim et al., 2015*; *Martin et al., 2016*). Various studies have consistently shown that oocytes from women over 40 years of age have decreased quality (*Alasmari, Son & Dahan, 2016*; *Hourvitz et al., 2009*); subsequently, the resultant embryos will show declining developmental potentials (*Marquez et al., 2000*). A large prospective study concerning the effect of age on perinatal livebirth outcomes varied by the number of embryo transferred showed that the risk of multiple births was lower among women older than 40 years (*Lawlor & Nelson, 2012*). Similar to previous studies, the data in this study suggests that women older than 40 years old have a reduced risk of

**Table 3 Clinical outcomes in double embryo transfer, stratified by age and prognosis.**

| | Favorable prognosis | Average prognosis | Poor prognosis | P-value[a] |
|---|---|---|---|---|
| **35–37 years** | | | | |
| No. of cycles | 487 | 483 | 164 | |
| No. of pregnancies | 240 | 198 | 59 | |
| Clinical pregnancy rate[b,d] | 49.3%[e] | 41.0%[e] | 36.0%[e] | 0.003 |
| Implantation rate[b,d] | 310/974 (31.8) | 249/966 (25.8) | 68/328 (20.7) | <0.001 |
| No. of twin pregnancies | 70 | 51 | 9 | |
| Twin pregnancy rate | 29.2% | 25.8% | 15.3% | 0.092 |
| Live birth rate[b,d] | 217 (44.6)[g,i] | 185 (38.3)[g] | 51 (31.1)[g] | 0.006 |
| **38–40 years** | | | | |
| No. of cycles | 221 | 262 | 98 | |
| No. of pregnancies | 97 | 100 | 29 | |
| Clinical pregnancy rate | 43.9% | 38.2% | 29.6%[f] | 0.051 |
| Implantation rate[c,d] | 123/442 (27.8) | 122/524 (23.3) | 32/196 (16.3) | 0.007 |
| No. of twin pregnancies | 26 | 22 | 3 | |
| Twin pregnancy rate | 26.8% | 22.0% | 10.3% | 0.173 |
| Live birth rate[d] | 86 (38.9) | 91 (34.7) | 24 (24.5)[h] | 0.044 |
| **>40 years** | | | | |
| No. of cycles | 104 | 161 | 58 | |
| No. of pregnancies | 37 | 48 | 8 | |
| Clinical pregnancy rate[c,d] | 35.6% | 29.8% | 13.8% | 0.012 |
| Implantation rate[c,d] | 43/208 (20.7) | 53/322 (16.5) | 8/116 (6.9) | 0.039 |
| No. of twin pregnancies | 6 | 5 | 0 | |
| Twin pregnancy rate | 16.2% | 10.4% | 0% | 0.553 |
| Live birth rate[c,d] | 32 (30.8) | 42 (26.1) | 6 (10.3) | 0.013 |
| P value associated with the age and clinical pregnancy rate in each prognostic category[a] | 0.031 | 0.041 | 0.007 | |
| P value associated with the age and twin pregnancy rate in each prognostic category[a] | 0.256 | 0.076 | 0.708 | |
| P value associated with the age and live birth rate in each prognostic category[a] | 0.025 | 0.019 | 0.007 | |

Notes:
[a] P-value for global comparison between the three groups.
[b] Significant comparison between women with favorable and average prognosis.
[c] Significant comparison between women with average and poor prognosis.
[d] Significant comparison between women with favorable and poor prognosis.
[e] Significant comparison in clinical pregnancy rate between women aged 35–37 years and >40 years.
[f] Significant comparison in clinical pregnancy rate between women aged 38–40 years and >40 years.
[g] Significant comparison in live birth rate between women aged 35–37 years and >40 years.
[h] Significant comparison in live birth rate between women aged 38–40 years and >40 years.
[i] Significant comparison in live birth rate between women aged 35–37 years and 38–40 years.

double embryo implantation when using their own eggs, irrelevant of the prognosis. Thus transfer of two embryos is applicable to women older than 40 years. However, in women aged 35–40 years with favorable and average prognosis, clinical pregnancy rates were good, but twin pregnancy rates were high enough to reach more than 20%, which is slightly higher than previously reported in women aged 36–39 years (16.6%)

(*Veleva et al., 2006*). It is clearly that in these women better selection is obviously necessary. Given that women at these age groups has an increased perinatal risk and may become particularly vulnerable to multiple pregnancy (*Davis et al., 2008*; *Ubaldi et al., 2015*), possible eSET should be offered to women aged 35–40 years with favorable and average prognosis to increase the safety of ART; however, further data will be needed to support this conclusion.

Vitrification, or the rapid-freezing of embryos, has been implemented in many IVF centers and has been shown in comparative trials to be associated with higher rates of post-thaw survival and intact embryos than slow-freezing method (*Debrock et al., 2015*). The risk for cycle cancelation due to cryopreservation damage was becoming extremely low. Several studies have shown predictors of implantation after transfer of slow-freezing embryos including: percent blastomere survival after thaw (*El-Toukhy et al., 2003*; *Kaser et al., 2013*) and survival rate of intact embryo (*Burns et al., 1999*; *El-Toukhy et al., 2003*). In the present study, the use of vitrification technology has resulted in embryo survival rate of 97.8% and intact embryo rate of 89.2% (shown in Table 2). The factors reported in previous studies such as survival rate of embryos and percent blastomere survival after thaw may not be related to embryo implantation in this study. Thus, vitrification may emerge as a method to optimize the safety and efficacy of assisted reproduction.

An important question remains: what are acceptable twin pregnancy rates in Chinese women older than 35 years? Since introduction of the two child policy, most couples desire more than one child (*Li & Deng, 2017*). For women 35 years of age and older, a twin pregnancy may be the only chance to conceive and deliver a second child. It has been shown in the literatures that a significant proportion of women with advanced age have a clear preference for twins and view a twin pregnancy as being more desirable than having no child, even if associated with adverse maternal outcomes (*Fiddelers et al., 2011*; *Van Loendersloot et al., 2013*). So, the clinical decision between eSET and DET should not be based solely on the characteristics of the treatment cycle, but also on the preference of the patients.

The study has several limitations. First, due to the retrospective nature of the study and nonspecific criteria for DET, there is a possibility of a selection bias. Also, this study applies to an unselected group of patients with aged ≥35 years, namely, those with vitrification and transfer of two cleavage stage embryos. Because of the limited data, it was not possible to distinguish between those patients who had DET due to clinical history and those who had DET because only two embryos were suitable for transfer. Additionally, in early 2012, in order to reduce the multiple pregnancy rates, we implemented a policy of the transfer of a maximum of two embryos. The 2013 ASRM guidelines (*Practice Committee of American Society for Reproductive Medicine, Practice Committee of Society for Assisted Reproductive Technology, 2013*) recommended limits on the number of embryos transferring as two cleavage-stage embryos in favorable prognosis and three embryos in all others. The recent ASRM statement (*Practice Committee of the American Society for Reproductive Medicine, Practice Committee of the Society for Assisted*

*Reproductive Technology, 2017*) was published in 2017 after our study had been terminated. Even one year prior to the 2013 ASRM recommendations, our policy was to reduce the multiple pregnancy rates in this study. Moreover, our criteria defined the prognostic profiles were in line with the ASRM (patient age, embryo quality and previous IVF cycle). Finally, the quality of embryos transferred was only assessed by morphological score, which is believed to a simple, less costly and generally accepted method, but may not be consistent with true embryo quality (*Sigalos, Triantafyllidou & Vlahos, 2016*). However, an important strength that enhances the validity of our data is the consistency of our clinical and laboratory protocols during the 4-year study period (e.g., endometrial preparation, ultrasound-guided embryo transfer, the protocols for embryo grading, embryo cryopreservation and thaw protocols).

## CONCLUSIONS

From these results, it was concluded that women with a favorable or average prognosis have a high risk of twin pregnancies. The finding can be used to guide future practice: that is, performing eSET in women with favorable or average prognosis and DET in women with a poor prognosis. We hope that these results will encourage physicians to better tailor patient older than 35 years or counsel patients on eSET, which reduce multiple pregnancy and associated complications. The novel aspect of our study is that we stratified our comparisons by prognosis, which may be useful in making decisions between eSET and DET for women of advanced reproductive age.

### Funding

This study was supported by Natural Science Foundation of Zhejiang Province (No. LY17H040002), the Sci&Tech Program Project of Zhejiang Province (2018C37126), the Health Sci&Tech Plan Project of Zhejiang Province (2019KY363), the Special Project for the Research Institutions of Zhejiang Province (C11920D-04) and the Zhejiang Provincial Program for the Cultivation of High-level Innovative Health talents. The funders had no role in study design, data collection and analysis, decision to publish, or preparation of the manuscript.

### Grant Disclosures

The following grant information was disclosed by the authors:
Natural Science Foundation of Zhejiang Province: LY17H040002.
Sci&Tech Program Project of Zhejiang Province: 2018C37126.
Health Sci&Tech Plan Project of Zhejiang Province: 2019KY363.
Special Project for the Research Institutions of Zhejiang Province: C11920D-04.
Zhejiang Provincial Program for the Cultivation of High-level Innovative Health talents.

### Competing Interests

The authors declare that they have no competing interests.

## Author Contributions

- Yamei Xue conceived and designed the experiments, performed the experiments, analyzed the data, prepared figures and/or tables, authored or reviewed drafts of the paper, and approved the final draft.
- Kun Li performed the experiments, analyzed the data, prepared figures and/or tables, and approved the final draft.
- Songying Zhang conceived and designed the experiments, analyzed the data, authored or reviewed drafts of the paper, and approved the final draft.

## Ethics

The following information was supplied relating to ethical approvals (i.e., approving body and any reference numbers):

Reproductive Medical Ethics Committee of Sir Run Run Shaw Hospital, College of Medicine, Zhejiang University granted Ethical approval to carry out the study within its facilities (Ethical Application Ref: SRRSHRMEC2017005).

## Data Availability

The raw data are available in the Supplemental Files.

## Supplemental Information

Supplemental information for this article can be found online at http://dx.doi.org/10.7717/peerj.8308#supplemental-information.

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
