# Peer review of "The effect on twinning rate of transferring double vitrified-warmed embryos in women of advanced reproductive age: a retrospective study"

_PeerJ, doi:10.7717/peerj.8308_

## Round 0.1 · original submission · Major Revisions

The manuscript was evaluated positively by three expert reviewers.
Nonetheless, several issues remain to be clarified.
Please respond to all of the comments by the reviewers.

·

Basic reporting

The paper is well written and clear, with a good languange property.

The background and literature review is sufficient, but due to the year of the data, the 2017 ASRM guidelines for the number of embryos to transfer were not cited.
At the same time, the policy of double embryo transfer in the population is not acceptable for every institution all over the world and this can be a big problem when you try to make your research interesting to different public.

The first table is not cited in the text, and it could be substitute in part the description of the prognosis criteria in the Subjects part.

Experimental design

the research question is clearly defined and interesting: reducing the twinning rate is a goal of every ART center.
This is a retrospective study with all the known problems.
Methods are clearly presented.

Validity of the findings

At the other hand the statistical approach is quite poor: the stratification process and the Chisquare analysis does not permit to fully considered the conclusions as statistically robust.
A review with a professional statistic is much needed.

Additional comments

The article is interesting and very clear, even with a quite poor statistical approach.
The only limitation is that the embryotransfer policy is based on criteria that are not accepted in other countries so the findings are not completely exportable in other contexts.

Reviewer 2 ·

Basic reporting

There are minor grammatical mistakes in some places.

1. In the last sentence in the Introduction section, "...the number of previous cycle ..." should be replaced by "...the number of previous cycles ..." and so on.

2. Line 138, "continuous normal distributed" should be replaced by "continuous normally distributed." Please go over the article to correct all such mistakes.

3. Line 208: "previously studies" should be "previous studies."

Experimental design

No comments.

Validity of the findings

1. There are several entries in the P-value column in Table 2 and Table 3 designated as "NS." This acronym should be explained in the statistical analysis section. Also, why were these p-values not computed?

2. You should report standard errors with the estimates of proportion (e.g., the clinical and twin pregnancy rates in Table 3).

3. The authors claim (line 235) that "From these results, it was concluded that women with a favorable or average prognosis who are 35-40 years have a high risk of twin pregnancies." The corresponding p-values in Table 3 are still reported as "NS."

Reviewer 3 ·

Basic reporting

Manuscript Review: The effect on twinning rate of transferring double vitrified warmed embryos in women of advanced reproductive age.

Summary

The study is a retrospective analysis of advanced reproductive aged women who underwent vitrified-warmed double embryo transfer (DET). The purpose of the study was to investigate the twin pregnancy rate in women of advanced reproductive age undergoing DET in order to determine which population would be better served for elective single embryo transfer (eSET). The patients were subdivided into age (35-37, 38-40, and >40), as well as prognosis based on embryo transfer history and grade of embryos to be transferred. All embryos transferred were at the Day 3 stage, and each patient only contributed one cycle. Clinical pregnancy rates were defined by the observation of a gestational sac with our without fetal heartbeat on ultrasound evaluation 35 days after embryo transfer. The number of sacs was taken as the number of successful implantations. The results showed that both patients over 40 and those patients with poor prognosis had < 20% probability of twin pregnancy rate which can be used as a guide for practices in determining who to recommend eSET vs DET.

Experimental design

Appropriate

Validity of the findings

Appears appropriate

Additional comments

In your methods, line 81-82, you mention all embryos transferred were Day 3. Is this standard for your practice? Do you culture any embryos to day 5? If you culture embryos to day 5, do you have any results for blastocyst DET in advanced reproductive age women? ASRM committee opinion Blastocyst Culture in clinically assisted reproduction recommends blastocyst transfer in good prognosis patients due to higher likelihood of live birth.

The aim of the manuscript is to assist practitioners in knowing which patients of advanced reproductive age are at risk for twin pregnancy and who to recommend eSET. Do you have any data showing the live birth rates for these patients? That would be another helpful statistic to enhance the data and help with stronger evidence to support the recommendation.

I commend the authors on the retrospective study with a large patient volume and extensive data set that was compiled over many years. The manuscript is written clearly and is unambiguous with professional English language used throughout. It is a well-defined research question, and the methods are described with sufficient detail and information to replicate. The tables have sufficient information and are easy to read. I also commend the authors on the novel aspect of stratifying the patient by prognosis as well as age. The author has listed many weaknesses, including transfer of two cleavage stage embryos. Given the increased nature of blastocyst being the preferred embryo stage of transfer (ASRM committee opinion Blastocyst Culture in clinically assisted reproduction), I feel the data would be more helpful to have information on clinical and twin pregnancy rates of the subset of patients in embryos transferred in blastocyst stage. If there is data on this information, it should be included.

---

## Round 0.2 · accepted · Accept

Reviewer 2 suggests few minor changes. Please check their comments while in production.

Thank you for your submission to PeerJ.

·

Basic reporting

no comment

Experimental design

In the new form, it is well improved.

Validity of the findings

no comment

Additional comments

I am really pleased to note all the efforts of the authors in order make the paper clear and understandable.
In my opinion, it should be accept in the actual form

Reviewer 2 ·

Basic reporting

No comment

Experimental design

No comment

Validity of the findings

No comment

Additional comments

Minor comment:
In lines 143-144, "... way analysis of variance (ANOVA) test and the nonparametric Kruskal-Wallis test depending on normal or abnormal distribution, respectively. ..."

The line " ... abnormal distribution ..." should be changed to "... non-normal distribution ..."

Reviewer 3 ·

Basic reporting

no comment

Experimental design

no comment

Validity of the findings

no comment